# A Ketogenic Diet Followed by Gradual Carbohydrate Reintroduction Restores Menstrual Cycles in Women with Polycystic Ovary Syndrome with Oligomenorrhea Independent of Body Weight Loss: Results from a Single-Center, One-Arm, Pilot Study

**DOI:** 10.3390/metabo14120691

**Published:** 2024-12-08

**Authors:** Rebecca Rossetti, Vittoria Strinati, Alessandra Caputi, Renata Risi, Maria Letizia Spizzichini, Alessandro Mondo, Lorenzo Spiniello, Carla Lubrano, Antonella Giancotti, Dario Tuccinardi, Lucio Gnessi, Mikiko Watanabe

**Affiliations:** 1Department of Experimental Medicine, Section of Medical Pathophysiology, Food Science and Endocrinology, Sapienza University of Rome, 00161 Rome, Italy; rebecca.rossetti@uniroma1.it (R.R.); renata.risi@uniroma1.it (R.R.); marialetizia.spizzichini@uniroma1.it (M.L.S.);; 2Department of Gynecological, Obstetrical, and Urological Sciences, Sapienza University, 00161 Rome, Italy; 3Research Unit of Endocrinology and Diabetes, Università Campus Bio-Medico Di Roma, 00128 Rome, Italy

**Keywords:** insulin resistance, hyperandrogenism, oligomenorrhea, amenorrhea, ovarian volume, echographic parameters, hirsutism, very low carbohydrate diet, ketosis

## Abstract

**Background/Objectives**: Polycystic ovary syndrome (PCOS) is a common endocrine disorder in women of fertile age. Some studies suggest that a ketogenic diet (KD) may have a role in treating PCOS. We aimed to demonstrate the long-term effectiveness of a KD in PCOS. **Methods**: Eighteen patients with PCOS phenotype A were enrolled: 28% were of normal weight, 28% were overweight, and 44% had obesity. All participants followed a KD without meal replacements for 45 days. After this period, patients underwent gradual carbohydrate reintroduction over 45 days, and thereafter healthy eating indications were given. Twelve patients completed the study. The patients were assessed at baseline and after 6 months. Anthropometric data, body composition, pelvic ultrasound, blood chemistry, hirsutism, and menstrual cycles frequency were recorded; **Results**: Besides improvement in anthropometric parameters, menstrual cycles (p 0.012), ovarian volume (p 0.029), FSH (p 0.05), LH (p 0.037), and progesterone (p 0.017) improved independently of weight or fat loss. However, testosterone and hirsutism improvements were influenced by weight and fat mass reduction. **Conclusions**: Our study showed that a KD followed by gradual carbohydrate reintroduction in PCOS has beneficial effects medium term, mostly independent of body weight loss, even in normal-weight women, suggesting that nutritional ketosis exerts beneficial effects per se.

## 1. Introduction

Polycystic ovary syndrome (PCOS) is a prevalent endocrine disorder affecting many women in their reproductive years. The syndrome is characterized by a range of symptoms, including anovulatory oligomenorrhea, hyperandrogenism, and polycystic ovaries [1]. The combination of the clinical features of the syndrome (hyperandrogenism, ovulatory dysfunction, and polycystic ovarian morphology) can result in four phenotypes: phenotype A (full-blown syndrome PCOS) includes hyperandrogenism (clinical or biochemical), ovulatory dysfunction, and polycystic ovaries; phenotype B (non-polycystic PCOS) includes hyperandrogenism and ovulatory dysfunction; phenotype C (ovulatory PCOS) includes hyperandrogenism and polycystic ovaries; phenotype D (non-hyperandrogenic PCOS) includes ovulatory dysfunction and polycystic ovaries [2] These clinical features often co-occur with metabolic conditions such as insulin resistance, dyslipidemia, and an elevated risk of developing type 2 diabetes [3]. In addition, a recent meta-analysis showed that the overall prevalence of PCOS in patients with T2DM was 21% [4].

Insulin resistance is recognized to be the key pathophysiological feature of PCOS, compensatory hyperinsulinemia is a significant contributor to the hyperandrogenism which is a common feature in insulin-resistant women with PCOS [5]. Increased serum insulin stimulates ovarian androgen production, but also reduces sex hormone binding globulin (SHBG) production in the liver further increasing serum levels of free bio-available androgens [6]. Underlying this syndrome appears to be a vicious circle whereby androgen excess promotes abdominal adipose tissue deposition and visceral adiposity inducing insulin resistance and compensatory hyperinsulinism, which further facilitates androgen secretion by the ovaries and adrenal glands in women with PCOS [7]. Insulin resistance and ovarian hyperandrogenism, which promote the accumulation of intra-abdominal fat, appear to be the primary determinants of the metabolic abnormalities present in women with PCOS [8]. Moreover, PCOS has long-term health implications, including a heightened risk for cardiovascular diseases, infertility, endometrial cancer, and associated mental health challenges such as depression and anxiety [9].

One of the critical aspects of PCOS that complicates its management is the heterogeneity in its presentation. While many women with PCOS struggle with weight issues, many do not have excess body weight. This is particularly relevant because most current therapeutic interventions for PCOS, including lifestyle modifications and pharmacological treatments, are geared towards restoring normal body weight [10]. As a result, these interventions may not be applicable or effective for PCOS patients without weight issues, leaving a gap in treatment options for this subgroup. Given the limitations of existing treatments that often focus on weight management, there is a pressing need for alternative therapeutic strategies to address the underlying issues of PCOS more comprehensively. The ketogenic diet (KD), initially developed for epilepsy management, has recently gained attention for its benefits in various metabolic conditions. These range from obesity and diabetes [11] to nonalcoholic fatty liver disease (NAFLD) [12], immunomodulation [13], and certain types of cancer [14]. Preliminary research suggests that a KD could offer benefits in managing PCOS, particularly in terms of improving insulin sensitivity and reducing body weight [15,16,17,18]. However, these studies have generally been short-term and have focused on very low-calorie ketogenic diets in women with obesity. As maintaining a KD long-term is challenging, establishing whether the reported benefits extend after the end of nutritional ketosis is crucial.

This study aimed to address this gap by exploring longer-term effects of an isocaloric or mildly hypocaloric KD on PCOS patients, regardless of their weight status. In doing so, it sought to offer a more nuanced understanding of the potential of KD as an alternative therapeutic strategy for PCOS, thereby setting the stage for more targeted and inclusive future research.

## 2. Materials and Methods

This prospective, longitudinal, single-arm pilot study was conducted at the Endocrinology and Gynecology units of Policlinico Universitario Umberto I in Rome. The study spanned a duration of 6 months, during which patients were evaluated at baseline and at the end of the 6-month period. Assessments at both time points included medical history, anthropometric measurements, body composition evaluation, gynecological examination, and biochemical analysis. Patients were selected from those attending the Endocrinology outpatient clinic at Policlinico Universitario Umberto I in Rome. Inclusion criteria were fertile age, diagnosis of PCOS according to the Rotterdam criteria, and oligomenorrhea defined as menstrual cycles longer than 35 days as a mean. Exclusion criteria were similar to those mentioned earlier, with the addition of patients who had not given informed consent. Pregnancy was excluded at baseline with a serum pregnancy test.

### 2.1. Dietary Protocol

All patients followed a KD for 45 days and a low-carbohydrate diet for the subsequent 45 days. Healthy eating indications were therefore to be followed thereafter. The KD involved a daily carbohydrate intake of less than 50 g and a daily protein intake of 1.3–1.4 g/kg, with the remaining caloric intake coming from fats. Calorie intake was dependent on baseline body weight: if the patient had a normal weight, she was prescribed an isocaloric diet based on calculated total energy expenditure; if she was overweight, she was prescribed a mildly hypocaloric approach (−300/500 kcal of TEE), with no patient being prescribed any less than a 1200 kcal diet. No meal replacements were recommended. Patients were advised to drink 2 L of water daily and were recommended vitamin and mineral supplements. During the first 45 days, patients were evaluated every three weeks in terms of capillary BHB, anthropometric, and adverse events. After 45 days, fruits and pulses were recommended as a sole source of carbohydrate, with other sources such as bread or rice to be consumed occasionally. Caloric intake was slightly decreased in those who were overweight at baseline by approximately 300 kcal, whereas normal weight women were instructed to reduce the amount of fats in their diet in order to maintain an isocaloric approach in favor of legumes and fruits. This gradual reintroduction of carbohydrates, to configure a low-carb diet, is part of classic ketogenic diet protocols and avoids the metabolically unfavorable effects of an excessively rapid reintroduction.

### 2.2. Data Collection

Data were collected at baseline before the KD and after 6 months. Anamnestic and anthropometric data included civil status, profession, education, comorbidities, pharmacological history, lifestyle, and weight history. Both at baseline and at 6 months, data related to PCOS symptoms were collected, including amenorrhea in the preceding 6 months, menstrual cycle frequency, and hirsutism (quantified via the Ferriman–Gallwey score). Blood pressure and heart rate were measured using a digital automatic device. Body composition was assessed using Dual-Energy X-ray Absorptiometry (DEXA). Various parameters were measured, including android and gynoid fat mass, total android and gynoid mass, and estimated visceral adipose tissue in terms of mass, volume, and area. Compliance was confirmed by the presence of capillary beta hydroxy butyrate levels > 0.5 mmol/L during all the checks carried out and tracked through automated measurement of blood beta-hydroxybutyrate (BOHB) concentrations using a specific device (GlucoMen Areo 2K, Menarini Diagnostics, Firenze, Italy). The development of side effects was investigated by the administration of semi-structured questionnaires during the checks carried out in the first six weeks. The questionnaires were aimed at investigating the presence of any symptoms, intensity and duration of these, and need for drug therapies for symptom management.

### 2.3. Ultrasound Parameters

Ultrasound parameters were measured using a Samsung ws80a ultrasound machine, equipped with a 7.5 MHz transvaginal probe and a 3.5 to 5.5 MHz transabdominal convex 3d probe. Transabdominal ultrasounds were performed with a full bladder, while transvaginal ultrasounds were performed with an empty bladder. Data collected included the volume obtained by a 3D reconstruction, diameters of both ovaries, and the morphological appearance of both ovaries [19].

### 2.4. Biochemical Analysis

Blood samples from fasting patients were collected between 8 and 9 in the morning. These samples were transferred to the local laboratory and treated following standard operating procedures. A range of biochemical analyses was performed. Complete blood count was analyzed using the ADVIA 2120i Hematology System (Siemens Healthcare, Erlangen, Germany). Glucose was measured using the hexokinase method (Cobas C 501 analyzer, Roche Diagnostics S.p.A., Monza, Italy; kit 04404483190). Urea (enzymatic colorimetric, kit 04460715190), creatinine (enzymatic colorimetric, kit 04810716190), total proteins (colorimetric, kit 03183734190), albumin (colorimetric, kit 03183688122), AST (colorimetric, kit 05850819190), ALT (colorimetric, kit 20764957322), total cholesterol (enzymatic colorimetric, kit 06380115119), HDL cholesterol (enzymatic colorimetric, kit 06380115119), and triglycerides (enzymatic colorimetric, kit 06380115119) were also measured using the Cobas C 501 analyzer (Roche Diagnostics S.p.A., Monza, Italy) with reagents supplied by Roche Diagnostics GmbH (Mannheim, Germany). LDL cholesterol was calculated using the Friedewald formula. Sodium (indirect potentiometric ion-selective electrode technique, kit 10825468 001) and potassium (indirect potentiometric ion-selective electrode technique, kit 10825441 001) were analyzed using the Cobas ISE 8000 analyzer (Roche Diagnostics, Monza, Italy). Insulin was measured using electrochemiluminescence (Roche Elecsys analyzer, kit 07027559188). TSH, FT3, FT4, estradiol, FSH, LH, progesterone, and testosterone were measured by chemiluminescent microparticle immunoassay (CMIA) on the Architect System (Abbott Laboratories, IL, USA) using the following kits: TSH 07K6225; FT3 07K6325; FT4 07K6529; estradiol 07K7225; FSH 07K7525; LH 02P4025; progesterone 07K7725; testosterone 02P1327. Cortisol was measured by radioimmunoassay using the Beckman Coulter analyzer (kit IM1841). All the analyses were performed according to local standard operating procedures. The HOMA index was calculated using the formula: HOMA-IR = (insulin (mU/L) × fasting glucose (mmol/L))/22.5. Due to the extreme irregularity of menstrual cycles, baseline assessment was not performed on a specific cycle phase.

### 2.5. Statistical Analysis

Statistical analysis was performed using SPSS 27.0 software. Variables were tested for normal distribution using the Shapiro–Wilk test. Continuous variables were expressed as means (standard error) when the variables were normally distributed and as medians (IQR) when the variables were not normally distributed. For the inferential statistics, a logarithmic transformation was performed to obtain an approximately Gaussian distribution. For the analysis of the outcomes, mixed generalized linear models were used, with random intercepts. Time was entered as a fixed effect. Correction for weight, fat mass, and HOMA-IR were applied for primary and secondary outcomes when appropriate. The sample size was calculated based on a 10% reduction in the prevalence of oligomenorrhea, chosen as the minimum clinically significant difference for improving reproductive health outcomes in this population, consistent with similar studies. With an alpha level of 0.05 and beta of 0.2, a sample size of 11 was determined. The baseline prevalence and anticipated reduction were based on prior studies on comparable populations undergoing similar dietary interventions. To account for an anticipated 40% dropout rate—estimated from historical data indicating higher dropout rates among normal-weight participants compared to those with obesity—18 participants were enrolled. This dropout estimate aligns with trends in longitudinal lifestyle intervention studies. Sensitivity analyses confirmed that the calculated sample size would maintain adequate statistical power despite variations in dropout rates or effect size.

## 3. Results

The study enrolled 18 patients through the gynecology and endocrinology units of Policlinico Umberto I in Rome from February 2021 to December 2022. In total, 28% had a normal weight, 28% were overweight, and 44% had obesity. All patients had PCOS phenotype A (full-blown syndrome PCOS: hyperandrogenism, ovulatory dysfunction, polycystic ovaries). None of the subjects were taking any drug therapy. Of these, six patients dropped out before the end of the study, four due to difficulty in adhering to the diet, and two patients due to personal reasons. Only those who completed the study were included in the analysis.

The anthropometric and biochemical characteristics of the study population included in the analysis are summarized in Table 1. Briefly, the average age was 26 ± 6 years, with an average BMI of 32 ± 6 kg/m^2^, an average waist circumference of 100 ± 12 cm, and an average HOMA-IR of 5.8 ± 3.7. The average cycle duration was 45 (IQR 25) days, the average Ferriman–Gallwey score was 20 (IQR 12), and the average total body fat percentage was 37.4 ± 6.2%. All patients had oligomenorrhea or amenorrhea, as per inclusion criteria.

### 3.1. Anthropometric and Metabolic Biochemical Changes

At six months, significant weight loss was observed (*p* = 0.003), along with a significant reduction in waist (*p* = 0.015) and hip circumferences (*p* = 0.001) (Figure 1A–D). Normal-weight patients changed their body weight by about 2 kg on average. There was also a reduction in the percentage of body fat (*p* = 0.003) and visceral fat (*p* = 0.037). Notably, lean mass was slightly but significantly reduced (*p* = 0.005). A trend towards improvement in insulin resistance (HOMA-IR, *p* = 0.076) was noted, and the lipid profile was improved, with HDL and triglycerides significantly increased and decreased, respectively (*p* = 0.004 and *p* = 0.002) (Table 1).

### 3.2. Safety Parameters

Renal function, complete blood count, protein electrophoresis, and electrolytes remained unchanged (*p* = ns). Despite the diet’s relatively high lipid content, no worsening in the lipid profile was observed. Interestingly, AST and ALT levels were both reduced (*p* = 0.006, *p* = 0.003), suggesting a reduction in the hepatic steatosis often associated with PCOS (Table 1). No significant adverse events were reported; some patients reported occasional mild constipation, lasting a few days, which was easily managed by increasing hydration and dietary fiber; no medication was needed. All patients reported good tolerability, and compliance was confirmed by the presence of capillary beta hydroxy butyrate levels > 0.5 mmol/L during the checks carried out (Figure 1).Beta hydroxy butyrate was positive only in measurements taken at week 3 (1.55 ± 0.9 mmol/L) and week 6 (1.25 ± 0.6 mmol/L), an expected result since the patients followed KD from week 1 to week 6 (Figure 1).

### 3.3. PCOS-Related Parameters

All patients reported regularization of menstrual cycle lengths, turning from a median of 45 (IQR 25) days to a median of 32 (IQR 6) days, *p* = 0.012), suggesting the restoration of ovulatory menstrual cycles (Figure 2). Ovarian volume significantly reduced, indicating an improvement in morphology and polycystic features (*p* = 0.029). Additionally, although the study was of limited duration, an improvement in hirsutism was observed (*p* = 0.037), as evidenced by the reduction in the Ferriman–Gallwey score post-dietary therapy (Table 2).

To confirm whether regularization of menstrual cycles was to be attributed to body weight or fat reduction, the model was adjusted for these parameters, showing this striking amelioration was indeed not dependent on anthropometric improvements. Similarly, adjusting for insulin resistance improvement showed that menstrual cycle improvement did not depend on it (Table 2).

By adjusting for the same parameters, it was shown that ovarian volume improvement was independent of weight or fat loss similar to menstrual cycle regularization, whereas it lost its significance upon HOMA-IR adjustment, suggesting that volume reduction was partially dependent on IR improvement.

Interestingly, hirsutism improvement, as shown by the Ferriman–Gallwey score reduction, was dependent on body weight and fat reduction, as well as on HOMA-IR improvement, suggesting that anthropometric and glucometabolic improvement may be the main drivers of hyperandrogenism amelioration (Table 2).

In terms of sexual hormones, FSH, LH, and progesterone showed significant changes with *p*-values of 0.050, 0.037, and 0.017, respectively. FSH, LH, and progesterone changes remained significant after adjusting for body weight and fat change, confirming that these changes did not depend on anthropometric improvements, as was for their direct clinical outcomes ovarian volume reduction and menstrual regularization. Conversely, testosterone reduction did indeed depend on body weight and fat mass reduction, similar to what observed for hirsutism improvement, suggesting that both biochemical and clinical hyperandrogenism are improved following a VLCKD due to its weight-lowering effect (Table 2).

## 4. Discussion

The therapeutic effects of the KD on various conditions, including polycystic ovary syndrome (PCOS), are increasingly being delineated in scientific literature. Preliminary studies have suggested positive outcomes in the treatment of PCOS in women with obesity [15,16,17,18]. Our study adds to this body of evidence by demonstrating significant improvements in PCOS-related outcomes as well as anthropometric characteristics following a KD in both lean and overweight women with PCOS. As stratification by BMI was not feasible given the small sample size and the small proportion of women with normal weight, statistics were adjusted by body mass loss in order to assess whether the PCOS improvements observed were dependent on weight loss, suggesting that this was not the case for most outcomes.

Specifically, the normal-weight and overweight patients lost an average of only 1.7 kg body weight, showing the same clinical benefits as the patients who showed greater weight loss.

We obtained excellent results concerning clinical parameters related to PCOS. Among the most relevant data, our study showed a resolution of menstrual irregularities, with a significant improvement in the frequency of menstrual cycles and a restoration of the cycle in cases of amenorrhea. This is in line with the study by Cincione et al., which, however, exclusively enrolled patients with obesity [16]. The fact that our study also included normal-weight patients and that the improvement maintained its significance after adjustment for body weight and fat loss, as well as after adjustment for IR improvement, suggests that the diet’s efficacy in treating PCOS is not dependent on its weight lowering effect. This is unsurprising thanks to two orders of reason. Low-grade inflammation lies at the base of all metabolic conditions as well as PCOS, and several studies showed that ketone body β-hydroxybutyrate (BHB) is capable of taming inflammation through several mechanisms such as inhibition of NLRP3 and some interleukins, likely also via microbiome modulation [20,21]. Interestingly, it was also recently reported both in vivo and in vitro that BHB reduces inflammation and cell apoptosis in ovarian cells, improving ovarian function in PCOS [22]. Moreover, BHB seems to exert some of its positive metabolic effects via SIRT1 activation, also without caloric restriction [23], further supporting the hypothesis that ketosis may favor PCOS improvement through several mechanisms that go beyond weight loss per se.

An innovative aspect of our study was the ultrasound assessment post-diet, which revealed a significant reduction in ovarian volume and improved morphological features and polycystic characteristics. Additionally, an improvement in hirsutism was reported, consistent with the existing literature [15]. These results suggest a therapeutic role for the KD in addressing ovulatory dysfunction, thus resulting in clinical improvement. Insulin receptors are present in normal and polycystic human ovaries and there is now an extensive body of evidence demonstrating the importance of the insulin signaling pathway in the control of ovulation [24,25,26]. Moreover, in PCOS, ovarian insulin action on steroidogenesis is preserved, despite resistance to insulin’s metabolic actions, suggesting that in PCOS there is selective insulin resistance in the ovary [27]. In the perspective of this selective resistance, the significant reduction in circulating insulin levels, despite the absence of a significant drop in HOMA IR, could partly justify the positive results obtained.

One finding to underline is the significant reduction in FSH. Physiologically, FSH is involved in both estrogen production, by stimulating aromatase production in granulosa cells, and ovarian follicle maturation. When a dominant follicle takes over, it secretes estradiol and inhibin, and FSH secretion is suppressed. When the dominant follicle produces enough estradiol to maintain levels of 200–300 pg/mL for 48 h, the hypothalamus responds with a surge of GnRH that stimulates the secretion of gonadotropic hormones instead of inhibiting them. FSH peaks at the same time as the increase in LH, which causes ovulation. FSH then remains low throughout the luteal phase, preventing new follicles from developing [28]. In PCOS, the LH/FSH ratio is skewed due to persistently rapid GnRH pulses. This skewed ratio leads in turn to the theca cells of the ovaries producing excess androgen while the granulosa cells do not produce enough aromatase to convert the androgens to estradiol, lacking the estradiol peak necessary for ovulation and the consequent drop in FSH [29]. The reduction in FSH values could therefore be due to an improvement in ovarian hyperandrogenism. Further studies are needed to explain whether the mechanism responsible is a simple reduction in the insulin action of ovarian aromatase [30,31,32].

Interestingly, biochemical and clinical hyperandrogenism were significantly improved, but this was dependent of body weight and fat mass loss. Indeed, visceral obesity contributes to the pathogenesis of PCOS by exacerbating insulin resistance and hyperandrogenism [8]. In this context, the KD proved extremely effective in weight loss and, consequently, could counteract the pathological mechanisms of the syndrome, improving its clinical aspects. Noteworthy, insulin resistance and hyperinsulinemia are also found in normal-weight PCOS patients, contributing to the syndrome’s etiopathogenesis [21].

From a metabolic standpoint, our study showed significant improvements in HDL cholesterol and triglycerides, with total and LDL cholesterol being unchanged. These results confirm that the KD does not worsen the lipid profile despite its relatively high lipid content. PCOS is often associated with non-alcoholic fatty liver disease (NAFLD), which is directly related to insulin resistance [33]. Preliminary studies suggest a slight improvement in liver function following a KD in patients with NAFLD [12]. Our study also explored liver function before and after the ketogenic diet, showing a reduction in ALT, AST, and GGT levels.

Our study has several limitations, such as the small sample size and the low percentage of normal-weight patients enrolled. Additionally, it is a single-arm study, so a major limitation is the lack of control group and the impossibility to make a comparison with other dietary and pharmacological treatments, so we can only assume, based on the evidence in the literature, the causal role of ketosis warranting the necessity of further research. It should also be noted that many patients had extremely irregular menstrual cycles at the time of enrolment, so blood samples were not always taken in the early follicular phase. Another aspect to underline is the need to extend the observation period in order to study long-term effects. Finally, as a limitation, the dropout rate must be taken into account, since the difficulty in adhering to the nutritional plan may make this approach unsuccessful for some patients. Nevertheless, dropout was a very rare phenomenon in patients with obesity, whereas the rate went up in normal-weight patients. This can presumably be attributed to the fact that in patients with weight gain, the benefit is associated with weight loss and therefore visible in the immediate period. Whereas normal-weight patients, not experiencing visible benefits immediately (hirsutism and menstrual irregularities take a few months to improve), are less likely to complete a dietary course that is in any case rigid and prolonged. However, the study also has strengths, including a 6-month follow-up, long enough to evaluate hirsutism improvement [34], ultrasound evaluations, and the inclusion of normal-weight patients to explore therapeutic effects in these. It should be noted that the enrolment of normal weight women was particularly challenging due to the scarce motivation of these individuals in undergoing a somewhat restrictive diet. This is a significant barrier to the application of such a treatment in these patients, despite its potential efficacy.

In conclusion, our findings indicate that the therapeutic effect of a KD in patients with PCOS may last months after the end of nutritional ketosis, concerning the syndrome’s anthropometric, clinical, and ultrasound aspects, even in normal-weight women. The treatment had no side effects and was well-tolerated, even more so supporting its potential as a valid alternative to symptomatic drug use in the therapeutic approach to PCOS. However, studies with larger sample sizes and the inclusion of a significant normal-weight sample in the study are required for a better understanding of the diet’s effects on PCOS.

## Figures and Tables

**Figure 1 metabolites-14-00691-f001:**
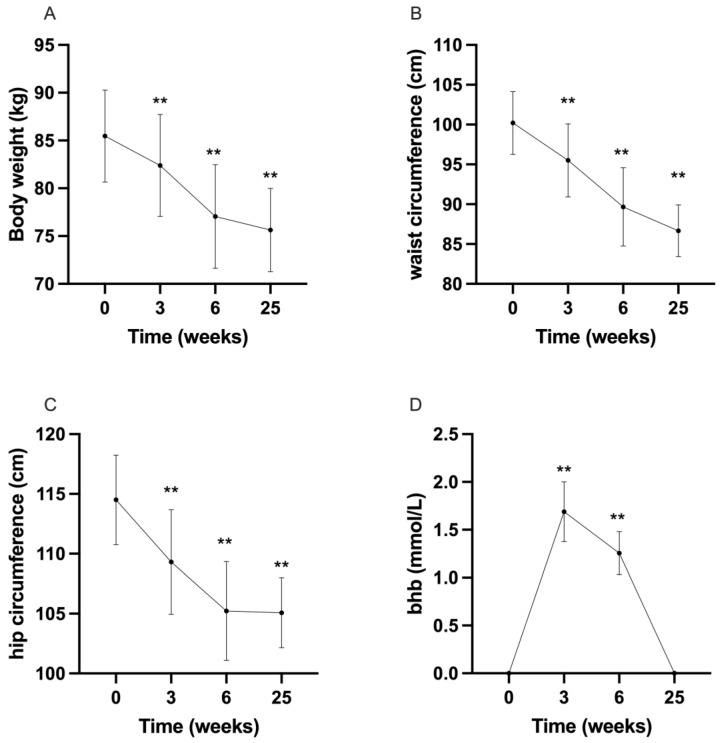
Changes in (**A**) body weight, (**B**) waist circumference, (**C**) hip circumference, (**D**) capillary beta-hydroxybutyrate (BHB) over time. ** *p* < 0.01.

**Figure 2 metabolites-14-00691-f002:**
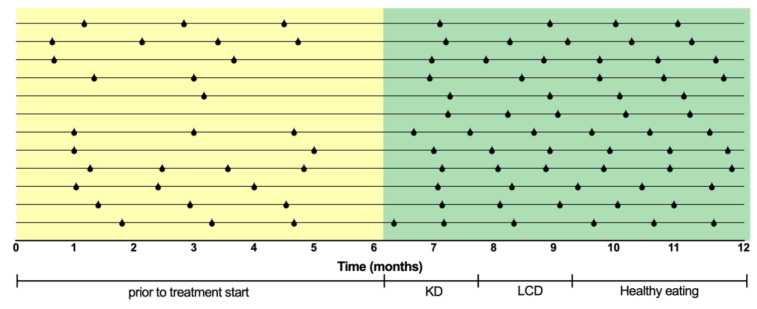
Menstrual cycle frequency of each patient who completed the study in the six months prior to study start and during the study up to the end of follow up. Each horizontal line represents one patient, each vertical line represents one bleeding. KD, ketogenic diet; LCD, low carbohydrate diet.

**Table 1 metabolites-14-00691-t001:** Anthropometric, metabolic efficacy, and safety outcomes.

	Before Diet	After Diet		p	p (Weight)	p	p
			% Change			(Fat Mass)	(HOMA-IR)
Number	12	12					
Age (years)	26 (6.15)	26 (6.15)					
BMI (Kg/m^2^)	32.18 (5.97)	28.71 (6.46)	−10.78%	0.008			
Weight (Kg)	85.46 (16.67)	75.63 (15.09)	−11.50%	0.003			
Waist Circumference (cm)	100.20 (12.47)	86.67 (11.26)	−13.50%	0.001			
Hip Circumference (cm)	114.50 (11.81)	105.92 (11.51)	−7.49%	0.024			
Fat Mass (Kg)	32.94 (9.69)	26.93 (9.53)	−18.25%	0.003			
Fat Free Mass (Kg)	50.73 (7.62)	46.89 (6.93)	−7.57%	0.005			
VAT mass (Kg)	0.37 (0.24)	0.26 (0.15)	−29.73%	0.037			
Glucose Metabolism							
Glucose (mg/dL)	84.08 (15.34)	82.00 (13.85)	−2.47%	0.424			
Insulin (µU/mL)	25.72 (14.06)	16.03 (15.59)	−37.67%	0.033			
HOMA-IR	5.81 (3.65)	3.51 (3.96)	−39.59%	0.076			
Cholesterol							
Total Cholesterol (mg/dL)	188.80 (34.70)	186.58 (27.02)	−5.98%	0.94			
LDL Cholesterol (mg/dL)	113.90 (34.15)	107.09 (23.7)	−30.70%	0.957			
HDL Cholesterol (mg/dL)	51.10 (16.25)	59.58 (17.07)	16.59%	0.004	0.02	0.016	0.004
Triglycerides (mg/dL)	118.20 (61.12)	81.91 (37.18)	−19.50%	0.022	0.204	0.259	
Liver Function							
AST (U/L)	22.36 (7.63)	18.00 (3.93)	−19.50%	0.006	0.105	0.117	
ALT (U/L)	28.18 (16.19)	17.42 (8.95)	−38.18%	0.003	0.127	0.105	
GGT (U/L)	16.91 (8.84)	11.67 (5.31)	−30.99%	0.008	0.216	0.195	
Safety							
Uric Acid (mg/dL)	25.75 (74.63)	27.85 (84.14)	8.16%	0.486			
Creatinin (mg/dL)	0.74 (0.10)	0.71 (0.10)	−4.05%	0.072			
Proteins (g/L)	72.00 (5.36)	73.42 (4.64)	1.97%	0.14			
Albumin (g/L)	47.00 (2.16)	48.17 (2.08)	2.49%	0.128			
Na (mmol/L)	139.40 (1.51)	139.67 (1.97)	0.19%	0.717			
K (mmol/L)	4.02 (0.34)	4.18 (0.30)	3.98%	0.248			
TSH T0(uU/mL)	1.85 (0.67)	1.84 (0.58)	−0.54%	0.904			
ft4 T0 (ng/dL)	1.06 (0.19)	1.12 (0.14)	5.66%	0.332			
ft3 T0 (pg/mL)	3.22 (0.38)	3.28 (0.86)	1.86%	0.742			

The variables are expressed as mean (standard deviation) and percentage change. The p is from a mixed generalized linear model, with random intercepts. Time has been entered as fixed effect. Correction for weight, fat mass, and HOMA-IR has been applied when appropriate. BMI, Body Mass Index; VAT, Visceral Adipose Tissue; HOMA-IR, Homeostatic Model Assessment for Insulin Resistance; AST, Aspartate Aminotransferase; ALT, Alanine Aminotransferase; GGT, Gamma-Glutamyl Transferase; Na, Sodium; K, Potassium; TSH, Thyroid-Stimulating Hormone; ft4, Free Thyroxine; ft3, Free Triiodothyronine.

**Table 2 metabolites-14-00691-t002:** Clinical and hormonal outcomes in regard to PCOS.

	Before Diet	After Diet	% Change	p	p (Weight)	p (Fat Mass)	p (HOMA-IR)	p (Testosterone)
Clinical PCOS Outcomes								
Menstrual cycles rhythm (days)	47.5 (25.2)	32 (5.9)	−32.63%	0.012	0.039	0.034	0.023	0.08
Mean Ovary Volume (cm^3^)	11.72 (6.60)	8.42 (5.80)	−28.16%	0.029	0.056	0.040	0.265	
Ferriman–Gallwey Score	20 (12)	16.5 (9)	−17.50%	0.037	0.168	0.134	0.099	
Sexual Hormones								
Estradiol (pg/mL)	132.75 (85.74)	97.16 (54.96)	−26.81%	0.218				
FSH (mU/mL)	5.19 (2.53)	3.65 (1.73)	−29.67%	0.050	0.018	0.046		
LH (mU/mL)	18.26 (19.90)	5.70 (3.12)	−68.78%	0.037	0.013	0.030		
Progesterone (ng/mL)	1.28 (2.76)	5.51 (5.61)	330.47%	0.017	0.018	0.025		
Testosterone (nmol/L)	1.39 (0.45)	0.97 (0.56)	−30.22%	0.048	0.085	0.075		

The variables are expressed as mean (standard deviation), with the exception of menstrual cycles rhythm, mean ovary volume and Ferriman–Gallwey score that are expressed as medians (IQR). The p is from a mixed generalized linear model, with random intercepts. Time has been entered as a fixed effect. Correction for weight, fat mass, and HOMA-IR has been applied when appropriate. PCOS, Polycystic Ovary Syndrome; IQR, Interquartile Range; FSH, Follicle-Stimulating Hormone; LH, Luteinizing Hormone; HOMA-IR, Homeostatic Model Assessment for Insulin Resistance.

## Data Availability

The data sets generated and/or analyzed during the current study are not publicly available but are available from the corresponding author on reasonable request.

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
