# Peer review of "A Ketogenic Diet Followed by Gradual Carbohydrate Reintroduction Restores Menstrual Cycles in Women with Polycystic Ovary Syndrome with Oligomenorrhea Independent of Body Weight Loss: Results from a Single-Center, One-Arm, Pilot Study"

_metabolites, 2024, doi:10.3390/metabo14120691_

Round 1
Reviewer 1 Report
Comments and Suggestions for Authors
The study was designed as a single-arm pilot study, meaning there was no control group. This raises the possibility that the results obtained may be due to factors other than the (KD). Especially in multifactorial conditions such as PCOS, it may be difficult to attribute the effect solely to the KD without a comparison group. For future studies, the inclusion of a control group is recommended.
Sample size calculation details must be included in the text. Only 18 patients were included in the study, 6 of whom did not complete the study. Therefore, the sample size limits the reliability of the results and reduces the statistical power. In addition, the low proportion of normal-weight patients makes subgroup analyses according to different body mass indexes difficult. Studies with a larger sample size will contribute to more generalizability of the results.
The study included a 6-month follow-up period. In a chronic condition such as PCOS, a longer follow-up period may be needed to see the long-term effects of dietary intervention. A longer follow-up period is recommended to understand the long-term effects and possible side effects.
Blood samples were not taken in a specific cycle phase due to the irregular menstrual cycles of the patients. This may make it difficult to interpret the results, as there may be phase-dependent fluctuations in hormone levels. In future studies, taking all blood samples in a specific cycle phase, such as the follicular phase, may contribute to greater reliability of the results.
The study's relatively high dropout rate raises questions about the long-term feasibility of the diet. Alternative motivation strategies to ensure participants' compliance with the diet or flexibility in the diet protocol may be considered.
Although only a few patients reported mild side effects such as constipation in the study, it is important to monitor the potential side effects of long-term KD more comprehensively. In future studies, more systematic reporting of side effects and evaluation of possible vitamin/mineral deficiencies may be useful.
minor editing
Author Response
The study was designed as a single-arm pilot study, meaning there was no control group. This raises the possibility that the results obtained may be due to factors other than the (KD). Especially in multifactorial conditions such as PCOS, it may be difficult to attribute the effect solely to the KD without a comparison group. For future studies, the inclusion of a control group is recommended.
Thank you for your feedback. We agree, the lack of a control group is a major limitation of our study as it does not allow us to associate the effects obtained with the ketogenic diet with certainty. In the future, we hope to be able to demonstrate this through an RCT. In the meantime, we have made this limitation more explicit in the text.
Sample size calculation details must be included in the text. Only 18 patients were included in the study, 6 of whom did not complete the study. Therefore, the sample size limits the reliability of the results and reduces the statistical power. In addition, the low proportion of normal-weight patients makes subgroup analyses according to different body mass indexes difficult. Studies with a larger sample size will contribute to more generalizability of the results.
We have expanded and made the sample size description clearer, thank you.
The study included a 6-month follow-up period. In a chronic condition such as PCOS, a longer follow-up period may be needed to see the long-term effects of dietary intervention. A longer follow-up period is recommended to understand the long-term effects and possible side effects.
Thank you for the observation. A longer observation period, even 1-2 years, would be very helpful to confirm the durability of the benefits. However, there are few studies in the literature that reevaluate patients even up to 6 months after the end of therapeutic intervention. Therefore, we believe that our data may be an interesting cue for designing longer-term studies.
Blood samples were not taken in a specific cycle phase due to the irregular menstrual cycles of the patients. This may make it difficult to interpret the results, as there may be phase-dependent fluctuations in hormone levels. In future studies, taking all blood samples in a specific cycle phase, such as the follicular phase, may contribute to greater reliability of the results.
Thank you for pointing this out. Unfortunately, some of our patients were in amenorrhea and the others had very irregular periods at baseline, so it was impractical to perform blood draws at a specific phase of the menstrual cycle. Since we did not have a baseline to compare with certainty, we deferred to performing blood draws at a specific time frame even at T1.
The study's relatively high dropout rate raises questions about the long-term feasibility of the diet. Alternative motivation strategies to ensure participants' compliance with the diet or flexibility in the diet protocol may be considered.
Your observation is much appreciated, thank you. However, in our population, drop out was a very rare phenomenon in patients with obesity, whereas the rate went up in normal weight patients. This can presumably be attributed to the fact that in patients with weight gain, the benefit is associated with weight loss and therefore visible in the immediate period. Whereas normal-weight patients, not experiencing visible benefits immediately (hirsutism and menstrual irregularities take a few months to improve), are less likely to complete a dietary course that is in any case rigid and prolonged. Poor compliance of normal-weight patients is indeed a limitation, so clarifying that it will take time to achieve benefits that could improve quality of life should be mandatory in the patient interview.
Although only a few patients reported mild side effects such as constipation in the study, it is important to monitor the potential side effects of long-term KD more comprehensively. In future studies, more systematic reporting of side effects and evaluation of possible vitamin/mineral deficiencies may be useful.
Thank you for your comment, we have taken care to expand the discussion regarding side effects and their recording in the text. However, when they occurred, they were mild and self-resolved in a few days only with lifestyle modifications.
Reviewer 2 Report
Comments and Suggestions for Authors
The aim of present study is to demonstrate the long-term effectiveness of ketogenic diet in women with PCOS. It’s really interesting, that authors have included the patients with normal weight which should have made it possible to assess the independent contribution of the ketogenic diet to changes in body composition, biochemical and hormonal parameters. Besides, I have some questions.
Which PCOS phenotype had each patient? According to inclusion criteria only patients with phenotype C couldn’t participate. Taking into account, that phenotype B is characterized by the absence of PCO and phenotype D is characterized by the absence of clinical hyperandrogenism, predominance of these phenotypes could affect on the study results.
Why data were collected only after 6 months? Patients followed a KD only for 45 days. It’s not enough for changing testosterone level or PCO morphology, but it can determine some biochemical changes. Also, it would be interesting to see during which period (KD, low-carbohydrate diet or healthy eating) patients had a more pronounced weight loss.
Did the authors record changes in body weight in patients with normal weight?
The equipment on which the all measurements were done needs to be clarified, including laboratory parameters.
The testosterone level was measured by ELISA or LS-MS/MS?
Author Response
The aim of present study is to demonstrate the long-term effectiveness of ketogenic diet in women with PCOS. It’s really interesting, that authors have included the patients with normal weight which should have made it possible to assess the independent contribution of the ketogenic diet to changes in body composition, biochemical and hormonal parameters. Besides, I have some questions.
Which PCOS phenotype had each patient? According to inclusion criteria only patients with phenotype C couldn’t participate. Taking into account, that phenotype B is characterized by the absence of PCO and phenotype D is characterized by the absence of clinical hyperandrogenism, predominance of these phenotypes could affect on the study results.
Excellent remark, thank you. All our patients belonged to phenotype A, we have specified this within the text.
Why data were collected only after 6 months? Patients followed a KD only for 45 days. It’s not enough for changing testosterone level or PCO morphology, but it can determine some biochemical changes. Also, it would be interesting to see during which period (KD, low-carbohydrate diet or healthy eating) patients had a more pronounced weight loss.
Thank you for your sharp observation. We evaluated patients after 6 months in order to demonstrate the durability of the results obtained over time, even months after discontinuation of the dietary regimen. Regarding the period (KD, low-carbohydrate diet or healthy eating) in which patients had more pronounced weight loss, we added a figure (Fig.1A).
Did the authors record changes in body weight in patients with normal weight?
Thank you for the question. Normal-weight patients changed their body weight by about 2 kg on average. we have made it explicit in the text now.
The equipment on which the all measurements were done needs to be clarified, including laboratory parameters.
Thank you, we added this in the text.
The testosterone level was measured by ELISA or LS-MS/MS?
Testosterone level was measured by Chemiluminescence Immunoassay (CLIA).
Reviewer 3 Report
Comments and Suggestions for Authors
TITLE: A ketogenic diet restores menstrual cycles in women with Poly-2 cystic Ovary Syndrome with oligomenorrhea independent of 3 body weight loss: results from a single center, one-arm, pilot 4 study
The main aim is to investigate the longer-term effects of an isocaloric or mildly hypocaloric KD on PCOS patients, regardless of their weight status.
Overall, the report has shown promising results, and I wish that the author would consider following minor changes.
General:
1. Abstract: Abstract is too wordy. Kindly reduce it and be concise about your thought delivery. Results should be shown with a statistical approach with p-values.
2. As KD and low-carb diets were applied in the same protocol, the conclusion of the abstract did not depict the overall findings of the present study and future requirements. Please fix it.
Introduction:
1. Author, please provide the data regarding prevalence of PCOS among type 2 diabetes subjects worldwide.
2. Kindly provide the rationale and novelty about this study, as few studies on ketogenic diet and T2DM disorders have been published previously, including a 1-year KD intervention.
3. Please mention the pathways that are involved in the progression of PCOS and leads/affected by obesity/T2DM in human subjects/clinical studies.
Method:
1. Kindly explain why both KD and low-carbohydrate diets were applied instead of one of them. The effect of the intervention stated by the author—kindly explain which diet was more responsible for it and why?
2. Medication score—please mention whether subjects were using medication to manage T2DM and PCOS. Kindly mention the medication score of subjects.
3. No ketone in serum/plasma mentioned in the manuscript. The author confirms if ketone was analysed before, after, and at the end of the protocol and reports it with statistical analysis.
4. How did the author confirm if diet induced nutritional ketosis and was responsible for stated health outcomes?
5. Please provide the references for the “Ultrasound Parameters” method.
6. How were blood glucose and other variables analyzed? Instruments /kits? Please mention it.
7. Please mention the catalog number of reagents/ kits /chemicals that were used in this manuscript.
Results:
1. Anthropogenic variables should be presented in a monthly/3-6-month basis line graph.
2. All tables: Results should be reported either in fold-changed or percentage format along with p-value.
3. Why are serum/urine ketone bodies not reported? Please explain how the author confirmed if nutritional ketosis was induced by diet intervention and all results mentioned in the manuscript were caused by diet intervention and not by medication taken by subjects.
4. Medication score should be mentioned in the method and results section.
5. Ketone analysis must be reported and please mention the correlation of ketone to

Author Response
The main aim is to investigate the longer-term effects of an isocaloric or mildly hypocaloric KD on PCOS patients, regardless of their weight status.
Overall, the report has shown promising results, and I wish that the author would consider following minor changes.
General:
1. Abstract: Abstract is too wordy. Kindly reduce it and be concise about your thought delivery. Results should be shown with a statistical approach with p-values.
Thank you for your comment, according to your comments, we have modified the abstract.
- As KD and low-carb diets were applied in the same protocol, the conclusion of the abstract did not depict the overall findings of the present study and future requirements. Please fix it.
Thank you for the suggestion, we modified the abstract including p values and reducing the word count.
Introduction:
- Author, please provide the data regarding prevalence of PCOS among type 2 diabetes subjects worldwide
This is a very interesting point, thank you, we included this data.
- Kindly provide the rationale and novelty about this study, as few studies on ketogenic diet and T2DM disorders have been published previously, including a 1-year KD intervention.
Thank you for your comment. it is undoubtedly true that the effect of ketogenic diet on T2DM is widely studied in the literature. However, this was not our aim, and we apologize for the lack of clarity. What we wanted to study was the effect of such diet on clinical PCOS, specifically on the more tangible manifestations, such as menstrual irregularities and clinical hyperandogenism. He hope to have now clarified this in the text.
- Please mention the pathways that are involved in the progression of PCOS and leads/affected by obesity/T2DM in human subjects/clinical studies.
Thank you for the helpful suggestion. We have taken steps to expand the description
Method:
- Kindly explain why both KD and low-carbohydrate diets were applied instead of one of them. The effect of the intervention stated by the author—kindly explain which diet was more responsible for it and why?
Thank you for your question. KD is an effective and well-tolerated nutritional protocol but cannot be followed for too long periods. Thus, at the end of the period of true ketosis, it is customary to proceed to a gradual reintroduction of carbohydrates, which is in fact assimilated to a low-carb diet, for longer or shorter periods, until the eventual complete reintroduction to configure a balanced Mediterranean diet. Therefore, the low-carb diet period can be considered part of a proper ketogenic nutrition protocol. In this regard, in any case, the greatest anthropometric changes were found in the actual ketosis phase. We proceeded to modify the text.
- Medication score—please mention whether subjects were using medication to manage T2DM and PCOS. Kindly mention the medication score of subjects.
Thank you for underlining this aspect. None of the subjects were taking any drug therapy, and none of the subjects had T2DM. We have made this more explicit in the text.
- No ketone in serum/plasma mentioned in the manuscript. The author confirms if ketone was analysed before, after, and at the end of the protocol and reports it with statistical analysis.
Thank you for your observation. We included in the results that ketone measurements demonstrated the effective presence of ketosis. However, ketonemia was performed in order to monitor dietary adherence. It was performed at week 0, 3, 6 and week 25. However, as expected, the ketonemia at 0 and 25 weeks was 0, as the patients were in ketosis only for the first 6 weeks.. Therefore, these values were not included in the statistical analysis because at t0 (baseline, before diet initiation) and at reassessment at 6 months (when the ketogenic diet had been discontinued for months) the patients were not in ketosis. However, we have included a graph (fig.1D) regarding the trend of ketonemia. Hopefully, this will make it clearer.
- How did the author confirm if diet induced nutritional ketosis and was responsible for stated health outcomes?
Thank you for the brilliant observation, the absence of the control group prevents us from saying with certainty that the effects obtained are indeed diet induced. We can only assume, based on the evidence in the literature the causal role of ketosis. The absence of the control group unfortunately represents our main limitation, however we hope that our results can serve as a basis for the design of new studies. We amplified the concept in the limitations sections.
- Please provide the references for the “Ultrasound Parameters” method.
We added in the text, thank you
- How were blood glucose and other variables analyzed? Instruments /kits? Please mention it.
We modified the section Materials and methods, Thank you.
- Please mention the catalog number of reagents/ kits /chemicals that were used in this manuscript.
We modified the section Materials and methods, Thank you.
Results:
- Anthropogenic variables should be presented in a monthly/3-6-month basis line graph.
Thank you for your suggestion, we added a figure.
- All tables: Results should be reported either in fold-changed or percentage format along with p-value.
Thank you for your suggestion, we modified the tables
- Why are serum/urine ketone bodies not reported? Please explain how the author confirmed if nutritional ketosis was induced by diet intervention and all results mentioned in the manuscript were caused by diet intervention and not by medication taken by subjects.
Ketones were not reported because at t0 (baseline, before diet initiation) and at reassessment at 6 months (when the ketogenic diet had been discontinued for months) the patients were not in ketosis. However, we included a graph (fig.1) inherent in the trend of ketonemia. We hope it is clearer that way. Subjects were not taking any medication as we clarified in materials and method thanks to your suggestion.
- Medication score should be mentioned in the method and results section.
Thank you for your comment, our patients were not taking any drug therapy. They also did not need any symptomatic therapy during the observation time. We have made this explicit in the text.
- Ketone analysis must be reported and please mention the correlation of ketone to
Ketone analysis was not included because at t0 (baseline, before diet initiation) and at reassessment at 6 months (when the ketogenic diet had been discontinued for months) the patients were not in ketosis, so the ketonemia was negative as expected. However, we included a graph (fig.1D) inherent in the trend of ketonemia. We hope it is clearer that way.
Round 2
Reviewer 1 Report
Comments and Suggestions for Authors
The article is sufficiently developed and ready for publication
Author Response
We thank the reviewer for the positive feedback and help in improving the manuscript.
Reviewer 2 Report
Comments and Suggestions for Authors
Authors clarified all notes except description of equipment for laboratory parameters. Now it's a clear methodology of laboratory measurements, but the information regarding equipment (name, manufacturer) and kits (name, manufacturer). This information could help compare the results of this study with the results of other similar studies.
Author Response
We thank the reviewer for the positive feedback and help in improving the manuscript. The manuscript has been updated with kit numbers and instruments.
Reviewer 3 Report
Comments and Suggestions for Authors
Thank you for editing and carefully crafting the manyscript after working on the comments. All queries and responses have been corrected with good understanding.
1. However, in Fig. 1 - there was no indication of statistical significance in either graph. Graphs were looking imcomplete without stats. Kindly consider these points and mention the statistical significance in the graphs of Fig 1.
2. The manuscript was brilliantly improved; nonetheless, it was regrettable that references were not written uniformly. For example, certain journal titles were written in full form, while others were written in short form; the first alphabet of the reference was capitalized, while the rest were in small caps. Kindly resolve this issue.

NA
Author Response
We thank the reviewer for the positive feedback and help in improving the manuscript. The graph and references have been updated.